# Treatment of Mantle Cell Lymphoma in the Frontline Setting: Are We Ready for a Risk-Adapted Approach?

**DOI:** 10.3390/jpm12071134

**Published:** 2022-07-13

**Authors:** Lindsay Hammons, Timothy S. Fenske

**Affiliations:** Department of Hematology & Oncology, Medical College of Wisconsin, Milwaukee, WI 53226, USA; lhammons@mcw.edu

**Keywords:** mantle cell lymphoma, non-Hodgkin lymphoma, minimal residual disease, risk-adapted therapy

## Abstract

Mantle cell lymphoma (MCL), a type of B-cell non-Hodgkin lymphoma characterized by the t(11;14)(q13q32) translocation, is a clinically heterogenous disease which can range from indolent to highly aggressive. Numerous prognostic factors have been identified, including blastoid histology, the Mantle Cell Lymphoma International Prognostic Index (MIPI) score, high proliferation index, p53 deletions and/or mutations, complex karyotype, minimal residual disease, and several others. However, using these prognostic factors to guide treatment selection has largely remained elusive. Given the heterogeneous behavior of this disease and varying patient characteristics, we suggest that the time has come for a more risk-adapted approach to this disease. In this article, we review the numerous prognostic factors that have been described for MCL, both at the time of diagnosis and following first-line treatment. We then propose a risk-adapted approach to first-line therapy for MCL, which would reserve intensive therapy for the highest risk patients and spare others excessive toxicity.

## 1. Introduction

Mantle cell lymphoma (MCL) is a subtype of B-cell non-Hodgkin lymphoma (NHL), identified by the translocation t(11;14)(q13q32), which results in an IGH/CCND1 fusion gene, leading to overexpression of cyclin D1. MCL comprises about 3–6% of total NHLs, and the incidence of MCL in the U.S. is rising [1]. After consensus criteria were established for the diagnosis of MCL in 1994, a series of retrospective studies were published in 1995–1997, showing a generally poor prognosis for MCL, with a median survival in the 3–4 year range [2,3,4,5]. However, in more recent studies, median survival exceeds 10 years [6,7,8,9,10] particularly in patients under age 65 who undergo “intensive” induction. Intensive induction typically refers to first-line regimens that incorporate rituximab, high-dose cytarabine and/or autologous hematopoietic cell transplantation (auto-HCT).

One of the challenges with MCL is the heterogeneity of the disease, ranging from an indolent form to the highly aggressive blastoid variant. Indolent cases can often be observed for several years without treatment; whereas those with blastoid features or p53 mutations typically follow a very aggressive clinical course and require aggressive therapy. Due to this clinical heterogeneity, combined with the typical patient being over age 60 (often with co-morbidities), MCL is a disease for which a risk-adapted approach to treatment would therefore be ideal. In this review, we will discuss prognostic models and factors that have been identified for MCL, and present an argument to use such prognostic factors to direct therapy.

Numerous prognostic factors have been identified for MCL. Some of these factors are determined at the time of diagnosis, others are determined after the completion of first-line therapy, and still others are determined at the time of relapse. Because this article is focusing on frontline therapy, we will discuss prognostic factors determined at diagnosis and at completion of first-line therapy.

## 2. Prognostic Factors Determined at Initial Diagnosis

### 2.1. Morphology

There are different morphologic variants of MCL: classic, pleomorphic, and blastoid. Among these the pleomorphic and, in particular, the blastoid variants have been associated with inferior prognosis [2,11,12,13,14], although this has not been seen in all studies [15,16,17]. Other adverse features such as high proliferation index are more commonly seen in cases with blastoid morphology [15].

### 2.2. MIPI Score

A number of clinical, molecular and histopathologic variables have been evaluated in MCL patients at the time of diagnosis. The most widely used prognostic tool in MCL is known as the Mantle Cell Lymphoma International Prognostic Index (MIPI)) [18]. The MIPI score incorporates age, white blood cell count, LDH, and performance status. In the original study describing the MIPI score, patients were stratified into low risk (median OS not reached), intermediate risk (median OS 51 months), and high risk (median OS 29 months) [18]. The MIPI score has been independently validated in several additional studies [18,19,20].

### 2.3. Proliferation Index

Multiple studies have corroborated the prognostic value of the proliferation index. This is typically assessed by the immunohistochemical stain Ki-67 or MIB-1. While various cutoff points have been used in different studies, a proliferation rate >30% has been shown to be associated with inferior survival in several studies [14,21,22]. 

### 2.4. MIPI-c Score

A revised version of the MIPI score combined the MIPI score along with proliferation index > 30%. This tool more effectively stratifies patients than the MIPI score alone [15]. Of note, other MIPI-based tools have also been developed, but have not yet validated (see Other Molecular Markers Under Investigation).

### 2.5. p53 Alterations

Mutations and deletions in the p53 gene have also shown prognostic as well as predictive value in patients with MCL. *TP53* ***deletions*** have been associated with poorer prognosis in multiple studies. For example, in an analysis of patients from the Nordic MCL2 and MCL3 trials, Eskelund et al. found that in those with *TP53* deletions had a median OS of 8 years (*p* = 0.002), PFS of 3.1 years (*p* = 0.03) and time to relapse of 3.1 years (*p* = 0.0006), compared to patients without a *TP53* deletion) [23]. Another study by Delfau-Larue et al. similarly found inferior outcomes associated with *TP53* deletions, whether high-dose cytarabine was included in treatment or not [24]. Whereas the above studies showed a statistically significant impact of *TP53* deletions, a study by Greiner et al. only showed a trend of worse median OS of those with p53 genomic deletions versus those with wild type *TP53* alleles (2.1 versus 3.1 years; *p* = 0.081) [25]. Furthermore, a separate study by Delfau-Laure et al. showed that the presence or absence of *TP53* deletions did not seem to impact OS [26]. 

Compared to mono-allelic *TP53* deletions, *TP53* ***mutations*** that result in dysfunctional proteins generally seem to confer a poorer prognosis. Because p53 is a tetrameric tumor suppressor protein, a loss-of-function mutation in one allele can cause general phenotypic loss-of-function of the final tetrameric p53 protein (sometimes referred to as a “dominant negative” effect). On the other hand, a heterozygous deletion still results in all wild type monomers (from the intact allele), and thus the final tumor suppressor protein remains functional. Multiple studies have shown a poorer median OS and PFS in those with *TP53* mutations [26,27]. Importantly, in the Nordic study from Eskelund et al., even in patients receiving intensive induction followed by auto-HCT, *TP53* mutations were associated with a much worse median OS of 1.8 years versus 12.7 years in *TP53* unmutated patients (*p* < 0.0001). Although more common in patients with high proliferation index, high MIPI score and blastoid morphology, p53 mutations still retained prognostic value on multivariate analysis [23]. 

One retrospective study found that, among MCL patients who underwent reduced-intensity allo-HCT, outcomes were similar whether TP53 mutations were present or not [28]. Therefore, *TP53* mutation status can be prognostic *and* predictive, and potentially identify novel treatment approaches more appropriate for this high-risk group.

### 2.6. IGHV Mutations

*IGHV* mutation status is a well-established prognostic factor in chronic lymphocytic leukemia; it has also been shown to influence prognosis in MCL in one study [29]. Nevertheless, most of the literature is descriptive, suggesting that *IGHV* gene mutations are associated with selective forces that lead to the development of MCL [30,31], and one study did not see a difference in survival based on *IGHV* mutation status [32].

### 2.7. Complex Karyotype

Complex karyotype (CK), generally defined as MCL with karyotypes consisting of three or more abnormalities [33], has been associated with inferior outcomes in multiple studies, such as a shorter treatment-free survival and PFS. In addition, CK was shown to be an independent predictor of poor OS in multiple studies [33,34]. Additionally, one study suggested the combination of CK with TP53 mutation conferred particularly poor OS and PFS [27]. In multivariable analysis with MIPI and proliferation index, only CK was associated with inferior OS. Of note, p53 mutation was not assessed in this study. Unfortunately, neither intensive induction nor auto-HCT in first remission appeared able to overcome the poor prognosis associated with CK [34]. Further limiting the application of CK for risk-assessment in MCL, is that routine metaphase karyotyping is often not performed on the diagnostic lymph node biopsy. Unlike some other molecular techniques, metaphase karyotyping requires fresh tissue and cannot be performed after the fact on archival specimens.

### 2.8. Gene Expression Profiling

It has been known since 2003 that gene expression profiling (GEP) on viable biopsy specimens can stratify MCL patients into groups with widely varying OS [35]. However, this technique requires immediate extraction of mRNA from the diagnostic specimen, or specific methods of preserving the specimen, that are not done in routine practice. 

Using the Nanostring technology, GEP can be applied to formalin-fixed paraffin embedded (FFPE) specimens, as are typically available for MCL patients in routine practice. Using this approach, Scott et al. developed the MCL-35 assay which consists of a 17-gene proliferation signature [36]. The MCL-35 assay stratified patients with MCL at diagnosis into high-risk, standard-risk and low-risk cohorts with OS of 1.1, 2.6 and 8.7 years, respectively, and independent of MIPI score (*p* < 0.001). This assay was then validated by the European MCL Network using 127 tissue samples from patients from the Nordic MCL2 and Nordic MCL3 trials, also showing that a high MCL-35 assay score was associated with poorer OS (*p* < 0.0001), even when adjusted for MIPI score and Ki-67 index [37]. Thus, the MCL-35 assay was found to add prognostic value above and beyond that of the MIPI score or proliferation index. Although in principle the MCL-35 assay could be applied in routine practice, this is not yet an approved test for clinical use. 

### 2.9. Other Molecular Markers under Investigation

Many other molecular aberrations have been studied as potential prognostic factors in MCL, most often in combination with other prognostic factors. Notably, Ferrero et al. used the presence of *KMT2D* (also known as *MLL2*) mutations and/or *TP53* mutations/deletions, combined with the MIPI-c score to develop a new prognostic index, named the MIPI-genetic (or MIPI-g) score [38]. First, the authors showed, using 186 patients from the Younger MCL cohort, that the presence of *KMT2D* mutations was associated with a worse OS (*p* = 0.002); the authors then went on to show that *KMT2D* plus *TP53* mutations/deletions cumulatively had a worse OS (*p* < 0.0001). Lastly, they showed that a high MIPI-c score with *KMT2D* and/or *TP53* mutations/deletions conferred a very poor OS (*p* < 0.0001). Separately, Delfau-Larue et al. showed that *CDKN2A* mutations in combination with *TP53* mutations/deletions had an additive negative effect on OS (*p* < 0.001) [24]. Additionally, Eskelund et al. showed that combinations of *TP53* deletions with other mutated genetic markers (*NOTCH1/2*, *CDKN2A* and/or *WHSC1* mutations) stratified patients into four risk categories with differing PFS (0.003) and OS (*p* = 0.0003) [23].

Other small studies investigating *CDKN2A, NOTCH1* and/or *NOTCH2* mutations, *MYC* overexpression, *KMT2B* mutations, *NSD2* mutations, *CCND1* mutations, and/or *ATM* mutations have also suggested poorer prognosis in patients with MCL, although these studies are limited [11,39,40,41,42,43,44]. In addition, microRNA-18b (miR-18b) at diagnosis was evaluated in the Nordic MCL2 trial, and then validated as a prognostic factor using the Nordic MCL3 trial [16,45,46].

Whereas some of these molecular markers look promising, in many cases they have not been independently validated in other studies. In addition, they typically are not captured in routine clinical practice, making them of limited utility at this time. However, with more widespread availability of genomic profiling assays that can be applied to FFPE tumor specimens, identifying such mutations is, in principle, feasible in practice now.

### 2.10. Imaging at Diagnosis

Studies have also correlated various parameters of ^18^Fluoro-deoxyglucose position-emission tomography (^18^FDG-PET) imaging with prognosis at initial MCL diagnosis. For example, the LyMa-PET project found that an SUVmax > 10.3 was correlated with a shorter PFS (HR 5.41, *p* < 0.001) and shorter OS (HR 6.32, *p* < 0.001) in young, previously untreated MCL patients who received intensive induction, auto-HCT and rituximab maintenance [47]. Furthermore, the combination of SUVmax > 10.3 (compared to ≤10.3) with high MIPI score was significantly associated with inferior PFS (*p* = 0.0027) and OS (*p* = 0.0002). This suggested that patients with high combined SUVmax/MIPI scores may represent a particularly high-risk group who may benefit from alternative therapy. Another study that used staging SUVmax showed that 5-year OS for those with SUVmax ≤ 5 was 87.7% compared to 34% in those with SUVmax > 5 (*p* = 0.01) [48]. Furthermore, due to the invasiveness of bone marrow and lymph node biopsies and excisions, Bodet-Milin et al. explored the use of ^18^FDG-PET alone for initial staging; in addition to showing the prognostic significance of staging SUVmax in MCL, this study also examined the combination of MIPI ≤ 2 or >2 and staging SUVmax ≤ 6 or > 6, showing that those with low MIPI/low SUVmax had a significantly better event free survival rate compared to high MIPI/low SUVmax or low MIPI/high SUVmax or high MIPI/high SUVmax (*p* = 0.004) [49]. In contrast, other studies have not found ^18^FDG-PET at diagnosis to be predictive of PFS or OS [50,51].

Because of variation in the SUVmax cutoff, variability in SUVmax measurement, and lack of consistent association with PFS or OS, baseline PET parameters are unlikely to be robust enough to help direct a risk-adapted treatment approach.

## 3. Prognostic Factors Determined Following Treatment

### 3.1. Imaging after Treatment

Imaging, mostly using ^18^FDG-PET scans, as discussed prior, has been suggested to have prognostic capabilities at different time points throughout the course of treatment in MCL patients, although the literature shows mixed results. The Nordic MCL3 trial used standardized response criteria from 1999 to determine PET positivity of patients with MCL after first-line (induction) therapy and before transplant; of 125 patients, about 14% of patients were PET-positive and had an inferior OS compared to PET-negative patients (*p* < 0.0001) [46]. Additionally, after four years, 72% of PET-positive patients in this study progressed and/or relapsed compared to 23% of PET-negative group (*p* = 0.017). Looking at the same post-induction/pre-auto-HCT time point, another study used an SUVmax cutoff of 5 to determine that the 5-year OS for those with SUVmax ≤ 5 was 87.7% compared to 34% in those with SUVmax > 5 (*p* = 0.01) [48]. Four additional studies found that FDG-PET after induction chemotherapy significantly correlated with poorer PFS [52,53,54].

Conversely, Kedmi et al. found that there were no differences in either 3-year OS (*p* = 0.5) or 3-year PFS (*p* = 0.4) between a positive and negative post-induction-pre-auto-HCT FDG-PET scans in 58 patients with MCL [50]. Hosein et al. also showed no significant correlation with 3-year OS (*p* = 0.07) or 3-year event-free survival (*p* = 0.16) in 56 patients with MCL at this same time point [51].

Determining prognosis using imaging at other time points, including mid-induction therapy and post-induction/post-auto-HCT, have less data. Multiple studies showed no correlation between mid-induction treatment imaging and OS or PFS [50,51,54]. Similarly, one study by Kedmi et al. showed that PET status post-induction/post-auto-HCT did not seem to predict 3-year OS nor 3-year PFS [50]. The LyMa-PET trial, mentioned above, looked at different percent changes in SUVmax between initial FDG-PET and end-of-treatment ^18^FDG-PET (ΔeotPET) and their respective correlation to survival; using a ΔeotPET of 90.88% post-induction-post-auto-HCT, PFS was significant (*p* = 0.0209) but OS was not [47].

Of interest, is that many cases of MCL will not be very PET avid. In some cases, even with easily seen lesions on CT, MCL may not be PET avid at all. One recent review suggested that, without more data, perhaps post-treatment FDG-PET scan data would be more useful in certain MCL subgroups, such as those with highly FDG-avid scans at diagnosis, or those with blastoid variant MCL (which generally is quite FDG-avid by virtue of high proliferation index) [55]. Because the large majority of patients achieve a PET-negative remission following induction, and due to conflicting results in the literature, post-treatment PET data is unlikely to be of great value in defining a robust risk-adapted therapy approach.

### 3.2. Minimal Residual Disease

Minimal residual disease (MRD), after either induction chemoimmunotherapy, or after induction therapy followed by auto-HCT consolidation, has been shown to be predictive of outcomes in MCL in several studies. In these studies, MRD has typically been measured either by nested PCR, or immunoglobulin high throughput sequencing (Ig-HTS), performed on either peripheral blood or bone marrow. Multi-parameter flow cytometry can also be utilized to measure MRD, if there are circulating lymphoma cells. Depending on the technique utilized, sensitivity ranges from 10^−4^ to 10^−6^ [56,57,58].

The CALGB 59909 trial showed a significant correlation of MRD to outcome after induction therapy; of 39 patients with MCL and MRD samples, 46% were MRD negative after induction, and MRD-negativity improved to 74% with one course of intensification [59]. Additionally, on 3-year follow-up multivariate analysis, MRD positivity following induction was shown to predict disease progression with a HR of 3.7 (*p* = 0.016); 3-year PFS was 82% in MRD-negative cohort versus 48% in the MRD-positive cohort. Pott et al. evaluated 259 patients with MCL pooled from European MCL Network randomized trials, the MCL Younger [7] (patients ≤ 65 years old) and the MCL Elderly [60] (patients > 60 years old) trials. Using 1 year and 2 year landmarks, MRD negativity post induction chemotherapy and pretransplant was associated with significantly improved PFS; the MCL Younger cohort 2-year remission rate was 94% for MRD-negative patients versus 74% for MRD-positive patients (*p* = 0.022) and the MCL Elderly cohort 2-year remission rate was 77% for MRD-negative patients versus 34% for MRD-positive patients (*p* = 0.021) [61]. Cowen et al. also showed the correlation between MRD negativity and superior PFS and OS in patients after induction therapy and before auto-HCT; 5-year median OS and PFS for MRD-negative patients was not reached (both PFS and OS) and that of the MRD-positive cohort was 3.01 years (HR of 4.04; *p* = 0.009) and 2.38 years (HR3.69; *p* = 0.002), respectively [62]. Gressin et al. showed that MRD status post-induction and prior to auto-HCT predicted for improved PFS (*p* < 0.0001) and OS (*p* < 0.0001); this was also shown for mid-treatment MRD as well [63]. In the Nordic MCL3 trial, MRD-negativity post-induction was associated with a trend toward improved 4-year PFS, although this did not meet statistical significance (*p* = 0.29) [46].

One study by Klener et al., focused on older patients who received alternating 3 cycles R-CHOP and 3 cycles R-cytarabine, without auto-HCT, but with rituximab maintenance. The authors found that MRD (post 3 cycles or post 6 cycles) was not predictive of either PFS or OS. The authors hypothesized that, perhaps rituximab maintenance overcame the negative prognosis typically associated with MRD-positivity after induction [64]. This conclusion, however, is not supported by ECOG 1411, in which all patients received rituximab maintenance (±lenalidomide); MRD-negativity (by Ig-HTS or flow cytometry) was associated with PFS benefit in this study [65].

MRD appears to be predictive of outcomes when assessed following auto-HCT as well. In a pooled analysis of two phase 2 trials in 88 untreated, transplant-eligible MCL patients treated with rituximab-bendamustine (RB) x3 cycles and R-cytarabine x3 cycles followed by auto-HCT, sustained MRD negativity post-transplant was associated with increased rates of durable remission [66]. Furthermore, the Nordic MCL3 trial showed that PFS was dramatically better in patients who were MRD-negative post-transplant (4-year PFS of 90% vs. 40%, *p* < 0.0001) [46].

In the prospective Nordic MCL2 trial, MRD was assessed after induction followed by auto-HCT in 160 patients with MCL; if patients had MRD after auto-HCT or developed molecular relapse, they were pre-emptively given rituximab, after which 92% again attained MRD-negativity. Median molecular PFS and clinical PFS were 1.5 and 3.7 years, respectively, and, at the end of the study, 33 out of 38 MRD-negative patients were still in first CR [67]. This approach of MRD-driven pre-emptive rituximab may have less relevance now that most patients receive maintenance rituximab following auto-HCT; however, it could potentially be a useful strategy after patients complete rituximab maintenance.

Although most of the above studies show an improvement in PFS associated with MRD-negativity, most did not show improved OS. However, a recent meta-analysis combined the results of seven studies evaluating MRD in MCL [68]. Wu et al. showed that when combining data from the seven studies, MRD negative status pre-transplant, post-transplant and post-induction all significantly predicted PFS (HR 0.90, HR 0.11, and HR 0.48, respectively) as well as OS (HR 0.47, HR 0.14, and HR 0.74, respectively) [68]. Therefore, it may require large datasets to demonstrate OS benefits associated with MRD-negativity.

### 3.3. Indolent MCL

It has been appreciated for over 10 years that a subset of MCL will follow an indolent course. Some, but not all, indolent MCL cases have a “CLL-like” presentation with primarily blood and marrow involvement, splenomegaly, and relatively little adenopathy (sometimes referred to as “non-nodal” MCL). Similar to the strategy used in other indolent B-cell NHLs, MCL patients who are asymptomatic, without cytopenias, significant splenomegaly or bulky adenopathy can often be observed as the initial management strategy. For example, one observational study found that while 95% of patients with nodal disease required immediate treatment, only 49% of patients with non-nodal disease did (*p* < 0.001); corresponding median OS was 30 months in the nodal group and 79 months in the non-nodal group (*p* = 0.005) [69].

Subsequent studies have also reported favorable outcomes in patients able to undergo initial observation [12,70,71,72]. Martin et al. showed that delayed treatment (defined as greater than three months) before initiation of first-line therapy was associated with improved OS, compared to those who started treatment immediately. [70]. Similarly, Shanmugasundaram et al. showed that those who deferred therapy had a superior 5-year OS of 83.6% versus 72.6% compared to those treated immediately at diagnosis (*p* = 0.03) [71]. Additionally, among patients who were initially observed, once treatment was required, intensive induction strategies failed to produce improved PFS (*p* = 0.93). In contrast, for patients who underwent immediate therapy, intensive induction strategies (such as auto-HCT) were associated with improved outcomes. It is therefore very helpful, when possible, to see if patients can undergo initial observation since documenting this can then allow justification of a less intensive first-line therapy when the patient does need to start treatment.

## 4. Moving from Prognostic Factors to Risk-Adapted Therapy

From the above discussion, it is clear that there are many prognostic factors in MCL (see Table 1 and Table 2). However, in order to move the field forward, it is crucial to now move beyond identifying high risk features, to developing risk-adapted therapies, using prognostic and/or predictive factors. Ideally such prognostic factors can be applied at the time of diagnosis, or during treatment, to direct patients down different treatment pathways. Unfortunately, there are no first-line therapy trials which have been completed, in which high-risk patients were randomized to standard therapy versus some novel or alternative approach, to see if outcomes can be improved for these high-risk patients. There is at least one ongoing trial which aims to accomplish that. For example, ECOG-ACRIN 4151 (EA4151; NCT03267433) is a large, multicenter, prospective randomized trial in which previously untreated MCL patients undergo standard chemo-immunotherapy induction followed by restaging including FDG-PET, bone marrow biopsy, and peripheral blood Ig-HTS. Patients who are in CR and MRD-negative are then randomized to auto-HCT followed by maintenance rituximab, versus maintenance rituximab alone. Randomization will be stratified by MIPI-c and intensive (high-dose cytarabine containing) versus non-intensive induction. The underlying concept is that patients in MRD-negative CR are in a deeper remission, and therefore may not benefit as much from consolidation with auto-HCT. Depending on the results of the trial, it may therefore eventually be possible to spare MRD-negative patients the additional potential toxicity of auto-HCT. This study is actively accruing in the U.S., with over 400 patients randomized to date.

Until we have randomized clinical trial data, we feel that a risk-adapted approach could still be implemented based on available data, and based on information that can be obtained in routine clinical practice. For example, identifying patients with indolent MCL is very important, since these patients can safely be treated with non-intensive induction once treatment is needed. On the other end of the risk spectrum, patients with mutated p53 have dismal outcomes with standard chemo-immunotherapy. It is reasonable to consider alternative/novel approaches for such patients. Examples of this could include combinations of novel agents (such as ibrutinib/venetoclax) [73], and/or early application of CAR T-cell therapy and/or allo-HCT. When available, these high-risk patients should be strongly encouraged to participate in clinical trials evaluating novel combinations of targeted agents or other promising agents such as bispecific T-cell engagers or cellular immunotherapies.

For the remaining patients (i.e., those who are neither indolent MCL nor p53 mutated), assessment post-induction is important. Although there is no modern randomized trial showing OS benefit with auto-HCT [6,7,8,9,10], given the improved outcomes associated with intensive induction and auto-HCT, those who remain PET-positive and/or MRD-positive are high risk and, we feel, should undergo auto-HCT followed by rituximab maintenance. For those who are in MRD-negative CR the decision-making is more difficult. At this time, we feel the standard of care for such patients remains auto-HCT, (assuming they are transplant eligible), followed by 3 years of rituximab maintenance; it would be premature to recommend omission of auto-HCT in MRD-negative patients, until we have the results of EA4151 to inform this. For patients with additional risk factors such as high MIPI-c, complex karyotype, or other molecular aberrations (such as *TP53*, *KMT2D*, *CDKN2A*, or miRNA) auto-HCT would be even more strongly recommended at this time. Figure 1 shows our proposed risk-adapted approach to first-line therapy of MCL. This is a “forward looking” approach; until ECOG-ACRIN 4151 is fully accrued, we recommend continued enrollment of patients on that study.

## 5. Conclusions

The prognosis of patients with MCL has improved since the disease was first described in the early 1980s. However, MCL remains a very clinically heterogeneous disease, with a significant subset of patients having high-risk disease and inferior outcomes. With the standardization and validation of prognostic and predictive factors, new diagnostic and treatment modalities, along with the availability of robust and highly sensitive MRD-testing and high throughput sequencing of commonly mutated genes in MCL, risk-adapted first-line therapy the management of MCL has moved from a theoretical concept, to something that can now be applied in practice.

## Figures and Tables

**Figure 1 jpm-12-01134-f001:**
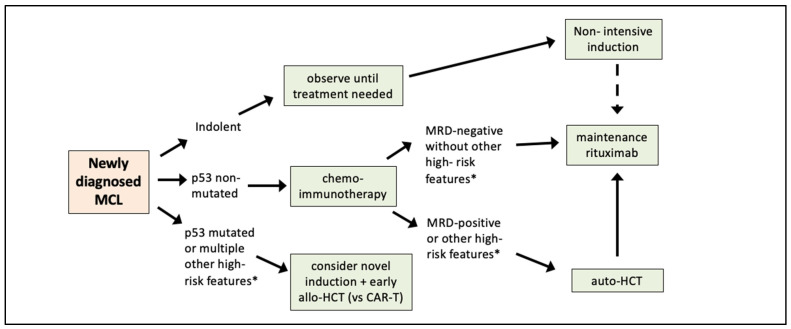
Proposed risk-adapted approach to first-line therapy of MCL. CAR-T = chimeric antigen receptor T-cells, auto-HCT = autologous hematopoietic stem cell transplant, MRD = measurable residual disease. * High-risk features include MIPI-c, complex karyotype, high risk genomic alterations (KMT2D, CDKN2A), and perhaps (in the future) MCL-35. Note: omission of auto-HCT based on MRD-negative status alone cannot be recommended until the results of ECOG-ACRIN 4151 are available.

**Table 1 jpm-12-01134-t001:** Prognostic Factors Determined at Initial Diagnosis.

	Poor Prognostic Factor	References	Outcomes (*p*-Value)	
			PFS or TTF	OS
Morphology	Blastoid histology *	[2]	-	<0.001
		[12]	-	<0.0001
		[13]	-	NS
		[14]	-	<0.001
		[15]	NS	NS
		[16]	NS	NS
MIPI Score	Intermediate- and high-risk score versus low-risk score	[18]	-	<0.001
		[19]	<0.001	<0.001
		[20]	-	<0.0001
Proliferation Index	Ki-67 or MIB-1	[14]	<0.001	<0.001
		[21]	≤ 0.030	-
		[22]	-	<0.001
MIPI-c Score	Intermediate and high-risk	[15]	<0.001	<0.001
p53 Alterations	p53 deletions	[24]	-	0.0003
		[25]	-	0.081
		[26]	-	0.7
		[23]	0.003	0.002
		[38]	<0.0001	<0.0001
	p53 mutations	[27]	<0.001	<0.001
		[23]	<0.0001	<0.0001
		[25]	-	0.0033
		[26]	-	0.0006
		[38]	<0.001	<0.001
IGHV Mutations	IGHV unmutated status	[29]	-	NS
		[32]	-	0.004
Complex Karyotype	≥3 karyotypic abnormalities	[33]	-	0.017
		[34]	<0.01	<0.01
		[27]	0.02	0.001
Gene Expression Profiling	MCL-35 assay	[36]	-	<0.01
		[37]	<0.0001	<0.0001
Other Molecular Markers	KMT2D mutation	[38]	<0.001	0.002
	MIPI-g	[38]	<0.0001	<0.0001
	CDKN2A mutation	[24]	-	0.0001
	CDKN2A mutation + TP53 mutation	[24]	-	<0.0001
	MYC over-expression (≥20%)	[40]	0.001	0.002
	NOTCH1 mutation	[42]	NS	0.002
		[43]	-	0.026
	NOTCH1 + NOTCH2 mutations	[43]	-	0.00034
	NOTCH2 mutation	[43]	-	0.00025
	microRNA 18b + MIPI-c	[45]	<0.001	<0.001
^18^FDG-PET	SUVmax > 10.3	[47]	<0.001	<0.001
	SUVmax > 5	[48]	<0.001	<0.01
	SUVmax > 6	[49]	-	NS
	SUVmax > 10.3 + high MIPI	[47]	0.0027	0.0002

PFS = progression-free survival, TTF = time to treatment failure, OS = overall survival, NS = not significant, MIPI = Mantle Cell Lymphoma International Prognostic Index, MIPI-c = MIPI along with proliferation index > 30%, MIPI-g = MIPI along with genetic factors (KMT2D mutation ± TP53 mutation/deletion), SUVmax = maximum SUV, auto-HCT = autologous hematopoietic stem cell transplant. * Blastoid or Blastoid/Pleomorphic morphology.

**Table 2 jpm-12-01134-t002:** Prognostic Factors Determined Following Treatment.

	Poor Prognostic Factor	References	Outcomes (*p*-Value)	
			PFS or TTF	OS
^18^FDG-PET	Mid-induction therapy	[50]	NS	NS
		[51]	NS	NS
		[54]	NS	NS
	Post-induction/before transplant	[46]	0.017	< 0.0001
		[52]	< 0.001	NS
		[53]	0.03	0.042
		[54]	0.001	NS
		[50]	NS	NS
		[51]	-	NS
	Post-induction/before transplant SUVmax >5	[48]	-	0.01
	Post-induction/post-auto-HCT	[50]	NS	NS
		[47]	0.0209	NS
Minimal Residual Disease	Post-induction/pre-auto-HCT	[59]	0.016	NS
		[61]	0.022	NS
		[61]	0.021	NS
		[62]	0.009	0.002
		[63]	< 0.0001	< 0.0001
		[46]	0.029	NS
		[68]	[HR 0.9, CI 0.84–0.97]	[HR 0.47, CI 0.31–0.72]
	Post-induction/on maintenance therapy	[64]	NS	NS
		[65]	0.002	-
	Post-induction/post-auto-HCT	[46]	0.0001	-
		[68]	[HR 0.11, CI 0.05–0.27]	[HR 0.14, CI 0.06–0.33]
Indolent MCL	Non-nodal disease vs. nodal disease	[69]	-	0.005
	Delayed treatment ≥ 3months	[12]	< 0.0001	< 0.0001
		[70]	-	0.0038
		[71]	-	0.03
		[72]	NS	NS

PFS = progression-free survival, TTF = time to treatment failure, OS = overall survival, NS = not significant, HR = hazard ratio, CI = 95% confidence interval, ΔeotPET = change in SUV from initial FDG-PET to end-of-treatment FDG-PET with cutoff of 90.88%, auto-HCT = autologous hematopoietic stem cell transplant.

## Data Availability

Not applicable.

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
