# Peer review of "Treatment of Mantle Cell Lymphoma in the Frontline Setting: Are We Ready for a Risk-Adapted Approach?"

_jpm, 2022, doi:10.3390/jpm12071134_

Round 1
Reviewer 1 Report
The paper is a reasonable summary of the role of RF in MCL management. The review is comprehensive and adjacent to the current medical knowledge. However I would suggest to change entirely figure 1. It does not reflex the current standard of care (neither ESMO nor NCCN), It does not reflex the current medical practice (ie by suggesting that only those with positive MRD or additional risk features would be the transplant candidates, which is evidently NOT true. I would suggest to change it, to the figure created by Prof Dreyling, after obtaining hos permission, which is much less controversial.
Author Response
Dear Reviewer 1,
We appreciate your comments and have taken these into consideration.
However, with regard to your suggestion that we change Figure 1, our understanding was that the purpose of the article was to propose a “risk-adapted“ approach to treating MCL. This requires a forward-looking approach. To simply recapitulate what is already in guidelines such as the NCCN guidelines would not accomplish this goal.
Figure 1 represents our suggestion of factors (including MRD) that could be used to guide treatment either currently or in the near future. We realize that this approach could change as new data emerges but is our best attempt at a forward-looking approach to incorporate such factors into treatment selection.
As a result, we would prefer not to revise Figure 1. We have, however made other minor changes as suggested by the reviewers (minor grammatical errors on page 2).
Thank you for the opportunity to publish in Journal of Personalized Medicine. We look forward to hearing back from you.
Sincerely,
Dr. Lindsay Hammons & Dr. Timothy Fenske
Reviewer 2 Report
This manuscript addresses an important and significant clinic question about the prognostic factors that have been identified for mantle cell lymphoma (MCL). MCL is a subtype of B-cell NHL with high heterogeneity, ranging from an indolent form to the highly aggressive blastoid variant. Authors summarized prognostic factors that have been identified for MCL and discussed their roles in MCL therapy. Overall, this review is interesting, and the manuscript is well written. Just has a typo indicated below:
1. In Page 2 “p53 Alterations” part, “xxxx Greiner et al only showed a trend in median OS of those of those with p53 genomic deletions xxxx”
Author Response
Thank you for the correction. Page 2 typo has been fixed.
~Lindsay
